# Temporality as Inductive Bias: The Relational Time Graph

## Abstract

Most temporal models treat time as discrete indices or elapsed intervals, leaving key properties of temporality only implicitly captured in hidden states. We introduce the Relational Time Graph (RTG), which provides an explicit inductive bias for temporality by learning a structured field of event-to-event gaps ($\hat{\Delta}$). RTG integrates three temporal properties into a unified energy formulation: relativity, where experiential gaps differ across individuals and contexts; co-existence, where private timelines are softly aligned through world-synchronization (WS) edges; and shock anchoring, where gradual change is preserved except at sudden surges in activity. We evaluate RTG on two behavioral datasets in commerce and news. On supervised prediction in e-commerce, RTG improves F1 over sequential baselines, with $\hat{\Delta}$ carrying the main signal and TV/WS priors stabilizing field formation. Overall, RTG provides measurable advantages on supervised tasks and consistent field formation on unlabeled data, demonstrating the value of making temporality explicit through a learnable $\hat{\Delta}$-field.

## 1 Introduction

Neural models of temporal data typically treat time as an auxiliary input. Sequence models use positional order, while elapsed intervals ($\Delta t$) are injected through encodings or features. Transformers rely on absolute, relative, or rotary positional schemes (Vaswani et al., 2017; Shaw et al., 2018; Su et al., 2024), which downstream recommenders adopt (e.g., SASRec (Kang & McAuley, 2018), BERT4Rec (Sun et al., 2019)) and extend with interval-aware attention (TiSASRec; (Li et al., 2020)). Temporal point processes such as RMTPP (Du et al., 2016) and Neural Hawkes (Mei & Eisner, 2017) parameterize event intensities as functions of $\Delta t$. Neural ODEs (Chen et al., 2018) and continuous-time graph extensions (Xhonneux et al., 2020; Poli et al., 2019; Guo et al., 2022; Behmanesh et al., 2023) evolve hidden states continuously. Generic encodings such as Time2Vec (Kazemi et al., 2019) and architectures such as the Temporal Fusion Transformer (Lim et al., 2021) also improve elapsed-time conditioning. Across these approaches, time enters as input or modulation, not as a structural object.

We focus on three properties of temporality that are central in human and social behavior but rarely modeled together: (1) relativity — heterogeneous temporal gaps across users or contexts (Koren, 2009; Sutton et al., 1999); (2) contemporaneity (world synchronization) — alignment of private timelines through shared anchors (Dasgupta et al., 2018; Trivedi et al., 2017); and (3) shocks — abrupt regime changes and bursts (Adams & MacKay, 2007; Killick et al., 2012; Kleinberg, 2002). Prior works address each of these in isolation. Our goal is to integrate them within a single framework, treating temporality not as elapsed indices but as an explicit structural bias.

We present the Relational Time Graph (RTG), which models temporality as a learnable $\hat{\Delta}$-field: event-to-event gaps defined on an interaction graph. Edge-level $\hat{\Delta}$ values are regularized by an energy functional combining three priors: semantic ordering (relativity), shock-aware smoothing, and world-synchronization. WS edges contribute no $\hat{\Delta}$ but act as anchors that softly align users experiencing the same items in the same time bucket. Shock priors down-weight smoothing near bursts, preserving discontinuities.

RTG does not replace elapsed-time conditioning; it adds a structural inductive bias that makes relativity, world synchronization, and shocks explicit. While distinct from prior mathematical treat-

ments, our perspective echoes related intuitions: shock-preserving PDE flows regard discontinuities as structural boundaries (Osher & Sethian, 1988), non-Euclidean embeddings capture relative scales across contexts (Nickel & Kiela, 2017), and geometric deep learning extends representation learning to manifold and graph domains beyond Euclidean time (Bronstein et al., 2017).

## 2  RELATIONAL TIME GRAPH

RTG represents temporality not as elapsed indices but as a field of event-to-event gaps defined on an interaction graph. The field is learned by minimizing a single energy that combines (when available) a supervised task loss with three priors: a semantic prior, shock-aware TV smoothing, and world synchronization (WS).

### 2.1  INTERACTION GRAPH AND WS EDGES

Let the interaction graph be $G = (V, E)$ with $V = U \cup I$. User–item transitions are edges $E_{ui} \subseteq U \times I \times B \times B \times T$, $e = (u, i, a \to b, t)$ where $B$ is the set of action types, $a$ is the previous action, and $b$ is the current action. Each edge corresponds to one concrete transition in a user's activity history (e.g., view→cart, view→purchase, or click→click in single-action logs). A learnable scalar $\hat{\Delta}_e$ is attached to every transition edge (see Section 2.2).

RTG encourages consistency across users who experienced the same item, same current action, and same external time bucket. We implement this by forming a line graph $L(G)$ whose nodes are the transition edges $E_{ui}$. Two nodes in $L(G)$ are connected if their underlying edges in $G$ satisfy the (item, current-action, bucket) condition. These connections are world-synchronization (WS) edges. They carry no $\hat{\Delta}$ values themselves. By default, WS edges act as anchors in the WS regularizer, softly aligning contemporaneous transitions without adding new predictive features. To keep training scalable, we sub-sample WS edges within each group rather than taking the full quadratic set.

### 2.2  $\hat{\Delta}$ FIELD

For each transition edge $e \in E_{ui}$, RTG predicts a scalar $\hat{\Delta}_e = f_\theta(a, b, u; \text{context}(u, i))$. This represents the perceived temporal gap of that transition, parameterized by the transition code $(a \to b)$, user identity, and lightweight context such as local history or item features. Learning is entirely edge-level; no node-level aggregates are used as training targets.

### 2.3  REGULARIZERS ON $L(G)$

We use a Huber TV penalty (Huber, 1992) with weights that suppress small noise but relax at large shocks: $\mathcal{E}_{\text{TV}} = \sum_{(e,e') \in E_L} w_{e,e'} \, \rho_\varepsilon(\delta_{e,e'})$ where $\rho_\varepsilon(\cdot)$ is the Huber loss, and $\delta_{e,e'} = |\hat{\Delta}_e - \hat{\Delta}_{e'}|$ for each WS-connected pair $(e, e') \in E_L$. The weight $w_{e,e'} = \min(w_e, w_{e'})$ is reduced when the underlying item–time bucket shows a strong burst, so sufficiently large shocks remain preserved as discontinuities. This design is consistent with change-point detection approaches that treat large deviations as structural boundaries rather than anomalies (Adams & MacKay, 2007; Killick et al., 2012; Aminikhanghahi & Cook, 2017; Altamirano et al., 2023). Note that shock-aware weighting is one instantiation showing that $\hat{\Delta}$-fields can selectively preserve socially meaningful bursts.

A gated quadratic penalty softly aligns contemporaneous transitions: $\mathcal{E}_{\text{WS}} = \sum_{(e,e') \in E_L} \mathbf{1}\{\delta_{e,e'} < \tau\} \, (\hat{\Delta}_e - \hat{\Delta}_{e'})^2$. This nudges $\hat{\Delta}$ values of co-clicking users closer together but does not add predictive signal. Related ideas appear in temporal knowledge graphs (e.g., HyTE (Dasgupta et al., 2018); Know-Evolve (Trivedi et al., 2017)) and session-based recommendation where co-click regularization is used to tie latent representations.

We bias $\hat{\Delta}$ toward semantic expectations. This is instantiated either as a margin ranking loss between transition codes (e.g., view→purchase > view→view), which we refer to as rank loss in experiments (see Section 4), or as a content-proximity regression that encourages $\hat{\Delta}$ to correlate with a semantic distance between consecutive items, computed from category, subcategory, or text

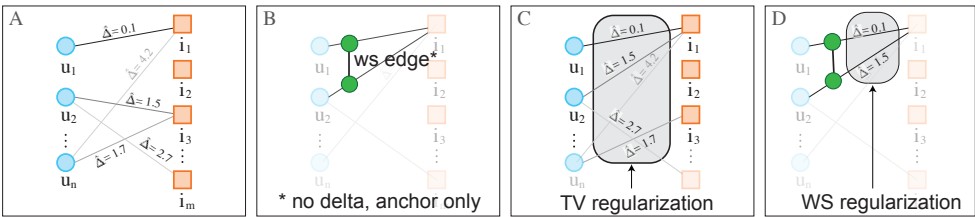

Figure 1: RTG components. (A) $\hat{\Delta}$ values on user–item edges, (B) world-synchronization edges, (C) shock-aware TV regularization, (D) WS regularization.

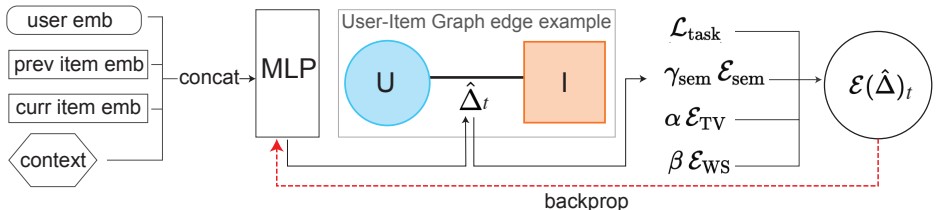

Figure 2: RTG training cycle. $\hat{\Delta}$ predictions on user–item edges are computed from embeddings via an MLP; task and regularization terms combine into the total energy $\mathcal{E}(\hat{\Delta})$, which backpropagates to update parameters and refreshes $\hat{\Delta}$ values on edges in the next epoch.

embeddings (see Section 4; (Wu et al., 2020)). Both play the same role as $\mathcal{E}_{\text{sem}}$, with the choice depending on the domain. An overview of these components is illustrated in Figure 1.

## 2.4 ENERGY FUNCTION

RTG minimizes a single energy:

$$\mathcal{E}(\hat{\Delta}) \;=\; \mathbf{1}_{\{\text{labels}\}}\,\mathcal{L}_{\text{task}}(\hat{\Delta}) + \gamma_{\text{sem}}\,\mathcal{E}_{\text{sem}}(\hat{\Delta}) + \alpha\,\mathcal{E}_{\text{TV}}(\hat{\Delta}) + \beta\,\mathcal{E}_{\text{WS}}(\hat{\Delta}). \tag{1}$$

**Task loss** $\mathcal{L}_{\text{task}}$ : used only when labels are available (e.g., binary cross-entropy for purchase prediction in Retail Rocket; see Section 4).

**Semantic prior** $\mathcal{E}_{\text{sem}}$ : instantiated as rank loss (Retail Rocket; see Section 4) or content regression (MIND; see Section 4).

**Shock-aware TV** $\mathcal{E}_{\text{TV}}$ : smooths small fluctuations, preserves sufficiently large shocks.

**WS regularizer** $\mathcal{E}_{\text{WS}}$ : softly aligns contemporaneous transitions, with no new features introduced. Note that the TV and WS terms act only as training-time regularizers; at evaluation, predictions depend solely on the learned $\hat{\Delta}$.

The overall RTG learning cycle is shown in Figure 2.

The resulting $\hat{\Delta}$ field is anchored by labels when available, ordered by semantic priors, smoothed except at shocks, and softly synchronized across co-occurring users.

## 3 RELATED WORK

### 3.1 TIME AS DYNAMICS

A major line of work represents temporal progress through continuous dynamics. Neural ODEs (Chen et al., 2018) replace discrete layers with a differential equation that evolves hidden

states continuously over depth or time. Graph extensions include Continuous Graph Neural Networks (Xhonneux et al., 2020) and Graph Neural ODEs (Poli et al., 2019), which replace discrete GNN layers with differential equations so that node embeddings evolve continuously. Continuous Temporal Graph Networks (Guo et al., 2022) extend neural ODE methods to event-driven temporal graphs. TIDE (Behmanesh et al., 2023) introduces a time-derivative term into diffusion-based message passing. These approaches model hidden state dynamics as a function of absolute time or elapsed intervals.

## 3.2 CHANGE-POINTS AND SMOOTHING

In cognitive science, salient change has been shown to anchor episodic memory, where discontinuities drive recall (Zacks et al., 2007; Davachi & DuBrow, 2015; Radvansky & Zacks, 2017). Analogously, our insight was as follows: in behavioral data, discontinuities or shocks can serve as structural markers that reorganize activity patterns and would also play a central role in analyzing and predicting future actions. In machine learning, abrupt shifts in temporal sequences have been studied through change-point detection, which aims to identify structural boundaries (Adams & MacKay, 2007; Killick et al., 2012; Aminikhanghahi & Cook, 2017). More recent scalable Bayesian approaches extend this line (Altamirano et al., 2023). In parallel, total variation denoising (Rudin et al., 1992) and fused lasso (Tibshirani et al., 2005) enforce piecewise smoothness while retaining sharp boundaries. Trend filtering (Kim et al., 2009) extends this principle, and graph signal processing work (Shuman et al., 2013) introduced graph total variation as a smoothness measure. Graph fused lasso (Padilla et al., 2018) applies TV regularization on graphs. Edge-preserving methods such as anisotropic diffusion (Perona & Malik, 2002), bilateral filtering (Tomasi & Manduchi, 1998), and adaptive TV approaches all reduce smoothing near strong discontinuities. RTG uses TV and Huber regularization but relaxes smoothing at shocks, preserving shocks as structural signals.

Prior work in continuous dynamics, change detection, and signal smoothing has generally modeled temporal structure through elapsed time or timestamp-based encodings. RTG takes a different approach by defining semantic change itself ($\hat{\Delta}$) as the temporal coordinate, and by regularizing it with TV and Huber while relaxing smoothing at shocks. This provides a structural representation of temporality that complements elapsed-time conditioning.

## 4 EXPERIMENTS

The experiments serve three purposes. First, we test whether RTG improves supervised prediction on temporal sequences when labels are available. Second, we examine whether the learned $\hat{\Delta}$-field forms a stable structure without collapsing. Third, we analyze the field in terms of distributions, semantic ordering, world synchronization, and shock preservation.

## 4.1 DATASETS

**Retail Rocket.** The Retail Rocket dataset (Retail Rocket, 2026) contains over 2.7 million e-commerce interactions with three event types: view, add-to-cart, and transaction. We segment sessions using a one-hour inactivity threshold and retain only users with at least three actions and items with at least five interactions. After filtering, the dataset contains about 232,000 sessions, 197,000 users, 79,900 items, and 1.26M events.

**MIND-Large.** The MIND dataset (Wu et al., 2020) contains large-scale news recommendation logs with impressions, clicks, and item metadata including category, subcategory, and title and abstract. Each row in the behaviors file corresponds to a user session with a single timestamp and a list of impressions and clicks, but does not include item-level timestamps. As a result, it is not possible to reconstruct item-level sequences with timestamp. Attempting supervised prediction under artificial ordering yields extremely low accuracy. Thus we do not perform supervised prediction tasks as we did with Retail Rocket, and instead focus on $\hat{\Delta}$-field analysis. To achieve this, we remove impression items to preserve item order as faithfully as possible. Each row then contains zero to a few clicked items, and we form sequences based on the row timestamp. When a row contains multiple clicks, all clicked items are extracted, and consecutive clicks are connected into transitions in the listed order, with all clicks assigned the same row-level timestamp, discretized into six-hour

buckets. One limitation is that the listed order within a row does not reflect the true chronological order, introducing noise. We accept this approximation because our goal in MIND is to study field formation.

For text features, we encode the concatenation of title and abstract using SBERT (all-MiniLM-L6-v2) (Reimers & Gurevych, 2019). Category and subcategory are also embedded with the same encoder. These embeddings are cached and projected to a fixed dimension for use in the $\hat{\Delta}$ predictor.

The MIND-Large dataset is reported to contain about 1M users and 160k news articles (Wu et al., 2020). In our preprocessed split, after filtering and sessionization, we obtain 711,000 users, 101,500 items, and 3.38M click interactions.

## 4.2 Experimental setup

**Retail Rocket.** We construct an interaction graph where nodes are users and items and edges represent transitions such as view to add-to-cart or add-to-cart to transaction. Actions are mapped to codes (1=view, 2=add-to-cart, 3=transaction). For each edge, we encode the previous and current action types as one-hot vectors, along with a shock weight. The supervised task is to predict whether the next action is a transaction.

The loss combines binary cross-entropy on transaction labels, a margin-based rank loss enforcing semantic ordering of transitions (e.g., 1→3 should be greater than 1→1), a Huber TV penalty with shock-aware weighting, and the WS regularizer. Training, validation, and test splits are created chronologically at a ratio of 70:15:15 based on session start time. Because the label is defined as the next action within each session, chronological splitting ensures there is no leakage of future information. World synchronization edges are built by grouping edges that share the same item, current action, and time bucket. The default time bucket is 24 hours, with robustness sweeps over various intervals reported in the results. We use a negative sampling ratio of one positive to two negatives, a batch size of 8,000 edges, a learning rate of $4 \times 10^{-4}$, and train for 150 epochs. Unless otherwise stated, experiments were run with random seed 42. Default regularizer weights were $(\alpha, \beta) = (0.1, 10.0)$.

**MIND-Large.** We construct an interaction graph $G = (U \cup I, E)$ from user–news clicks. Each news item is encoded with an ID embedding, category and subcategory embeddings, and a projected SBERT text embedding of its title and abstract. Each user has an ID embedding and a dynamic history of at most five recent clicks. A user context vector is obtained by attention pooling over the recent history. The input to the $\hat{\Delta}$ predictor is the concatenation of the previous-item and current-item embeddings with the user context vector.

Training minimizes the unsupervised RTG energy on MIND, combining (i) a category-hint loss term that encourages $\hat{\Delta}$ to increase with semantic distance between consecutive items, (ii) a Huber-TV smoothing term, and (iii) WS alignment. We additionally apply shock-aware weighting to the TV term: for each (item, time bucket) we compute a bucket-level score by comparing the observed click count to a rolling mean and variance (z-score), and down-weight the TV penalty near strong bursts to avoid oversmoothing. WS edges are constructed analogously to Retail Rocket, except that events are all clicks and grouping is by (item, time bucket). We train with batch size 512, learning rate $10^{-3}$, for up to 15 epochs, and report results at epoch 7 where the $\hat{\Delta}$ field stabilizes. Unless otherwise stated, experiments were run with random seed 42. Default regularizer weights were $(\alpha, \beta) = (0.1, 0.1)$.

## 4.3 Baselines

We compare RTG against standard sequential baselines on Retail Rocket. The baselines are LSTM (Hochreiter & Schmidhuber, 1997), SASRec (Kang & McAuley, 2018), and a multilayer perceptron. LSTM provides a recurrent sequential baseline, SASRec represents transformer-based sequential recommendation with positional encoding, and MLP provides a non-temporal feedforward reference. All baselines are trained and evaluated on the same data splits as RTG.

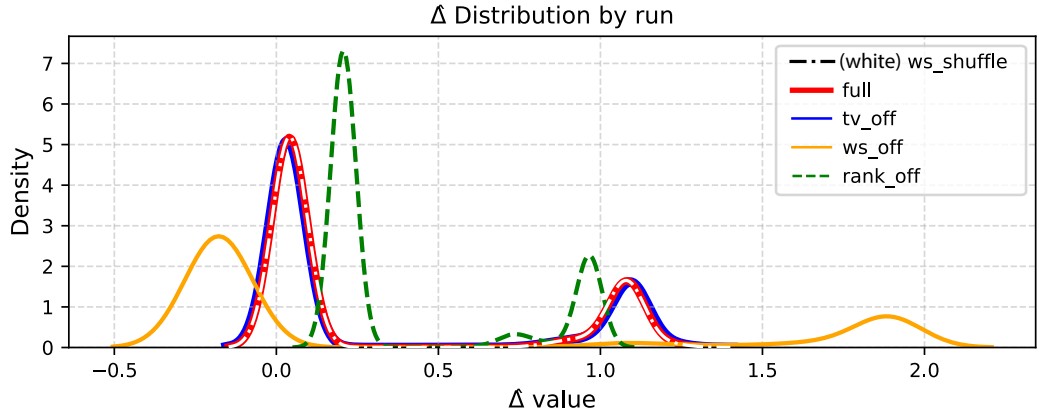

Figure 3: Distributions of $\hat{\Delta}$ under ablations on Retail Rocket. Kernel density estimates are plotted for Full RTG, WS-off, WS-shuffle, TV-off, and Rank-off.

## 5 RESULTS

Our primary question is not whether RTG achieves state-of-the-art accuracy, but whether it consistently learns a structured $\hat{\Delta}$-field that reflects temporality as an inductive bias. We therefore begin by analyzing field formation across domains (Section 5.1), examining stability, relativity, synchronization, and shock preservation. Supervised prediction is reported only as a secondary check to confirm that the learned $\hat{\Delta}$-field carries predictive signal (Section 5.2).

### 5.1 FIELD FORMATION RESULTS

#### 5.1.1 RETAIL ROCKET

Figure 3 shows the distributions of $\hat{\Delta}$ under different ablations. The full RTG produces a bimodal distribution: a sharp peak near 0.0 corresponding to transitions into *view*, and a broader secondary peak near 1.0 corresponding to transitions into *add-to-cart* or *purchase*. This interpretation is supported by Figure 15 (in the Appendix due to space constraints), which shows mean $\hat{\Delta}$ values grouped by transition type. WS-shuffle yields a nearly identical shape, indicating that the main effect of WS edges comes from grouping edges by item and time bucket. Even when user assignments are randomized, these groups still impose a weak smoothing constraint. By contrast, WS-off produces the broadest spread, showing that removing synchronization edges reduces cross-user alignment. TV-off has little effect on the global distribution, since TV regularization operates only locally on neighboring edges. Rank-off shifts the distribution into a narrow peak, with less spread across the range.

To examine user-level heterogeneity, Figure 4 plots per user $\hat{\Delta}$ histograms for three randomly sampled users (each with at least 20 edges) under full RTG. The distributions differ in location and shape across users: some concentrate near small gaps, others peak at larger values with different spreads, indicating that RTG learns user-specific event gap scales rather than a single global pattern. This illustrates the relativity aspect of the $\hat{\Delta}$ field.

Figure 5 shows the evolution of TV loss and relative TV-energy over epochs. The full RTG, WS-shuffle and $\hat{\Delta}$-only produce nearly overlapping curves that steadily decrease and plateau, showing that the field converges to a piecewise-smooth structure regardless of whether WS edges are shuffled. By contrast, WS-off remains at high relative TV-energy, indicating that WS is the main contributor to field stability.

We also vary the WS bucket size—the external time window used to group users into world-synchronization edges—from 60 minutes to 4320 minutes to test sensitivity to clock resolution. Relative TV-energy curves largely overlap across bucket sizes (Figure 6). Smaller buckets converge

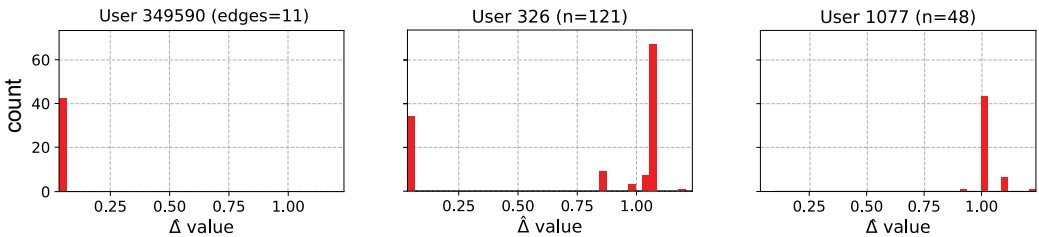

Figure 4: Per-user $\hat{\Delta}$ distributions for three randomly selected users ($n \geq 20$ edges) under Full RTG. Modes and spreads vary by user, showing user-specific event-gap scales.

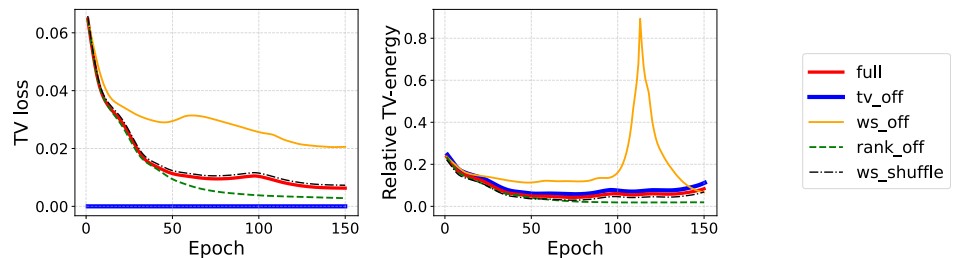

Figure 5: TV loss (left) and relative TV-energy (right) on Retail Rocket under ablations. Curves for Full RTG, WS-shuffle, and $\hat{\Delta}$-only nearly overlap, while WS-off remains high.

slightly faster, but by the end of training (150 epochs) the energy levels are similar. This result shows that $\hat{\Delta}$-field formation in RTG is stable across WS bucket sizes, suggesting robustness to reasonable variations in how external time is discretized.

### 5.1.2 MIND

Figures 7–8 illustrate how RTG forms a $\hat{\Delta}$-field on the MIND dataset under different ablations. The overall pattern mirrors Retail Rocket, though the relative roles of TV and WS differ across domains.

We begin with the category-loss-only setting (dashdot blue): the resulting $\hat{\Delta}$ distribution is broad and fragmented, with a shallow peak near 5 and mass spread across the range. Adding WS alone (green) does not fundamentally change the distribution, but slightly increases the central peak, suggesting that synchronization edges act as weak anchors on their own. In contrast, adding TV regularization (orange) produces a clear tightening effect: the distribution compresses into the 0–3 range with a sharp peak near 3, indicating that the field now carries distinct structure. Finally, combining TV and WS (red) yields the most stable field, with an even more concentrated peak than TV alone.

These results highlight that, on MIND, TV plays the dominant role in shaping the $\hat{\Delta}$-field, while WS provides additional but smaller alignment. This stands in contrast to Retail Rocket, where WS was the main driver of tightening and TV acted more as a local smoother. Across both datasets, however, the full RTG consistently produces the most stable and peaked field relative to its ablations.

Per-user $\hat{\Delta}$ histograms on MIND (in the Appendix due to space constraints, Appendix C) show similar relativity effects as in Retail Rocket: different users align to different gap scales, but RTG structures them into coherent bands.

Figure 8 shows field-level TV energy across epochs. TV energy decreases steadily when the TV prior is active, as minimizing TV directly reduces variance across adjacent edges. The full model achieves the lowest levels, while ablations without TV plateau. This indicates that TV is the primary driver of stabilizing the field on MIND, with WS providing smaller but consistent gains.

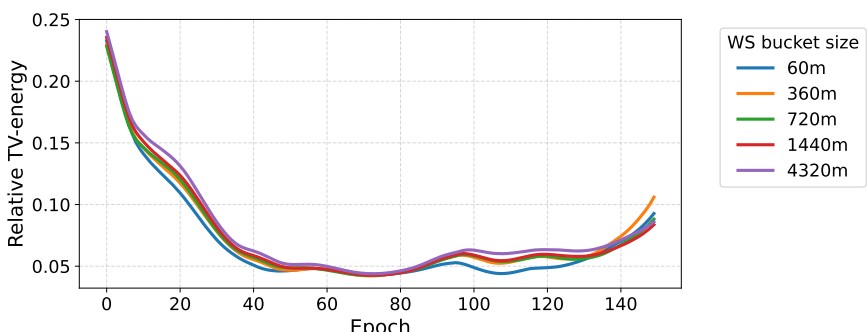

Figure 6: Effect of WS bucket size on Retail Rocket. Relative TV-energy over epochs for bucket sizes from 60 minutes to 4320 minutes. Curves converge to similar levels, indicating robustness to bucket size choice.

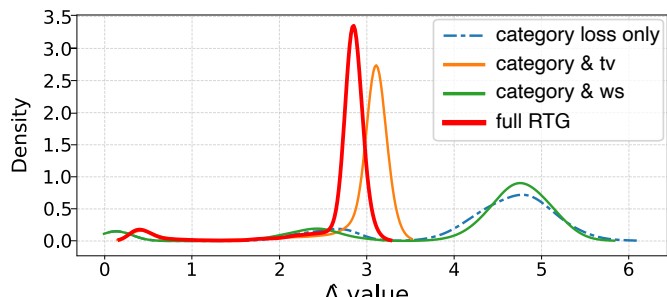

Figure 7: $\hat{\Delta}$ distributions under ablations (MIND). With category loss only (blue), the distribution is broad and shallow; adding ws loss (green) slightly increases the peak; using tv loss and category loss together (orange) sharply tightens the distribution; and the full rtg (red) yields the most stable field.

## 5.2 SUPERVISED LEARNING

We next evaluate RTG on supervised prediction in Retail Rocket. Table 1 reports results over three random seeds. RTG outperforms standard sequential baselines (LSTM, SASRec) and a feedforward MLP. Absolute F1 gains are modest, but they demonstrate that the learned $\hat{\Delta}$-field carries predictive signal in addition to forming a stable temporal structure. Detailed ablation studies, including the effect of individual regularizers, are reported in the Appendix.

## 6 DISCUSSION

### 6.1 RTG AS STRUCTURAL INDUCTIVE BIAS

Our experiments were not aimed at state-of-the-art benchmarks, but at a basic question: can a $\hat{\Delta}$-field be learned, smoothed, and aligned across users as an explicit structure for temporality? The results indicate that this is feasible. RTG consistently produces a piecewise-smooth $\hat{\Delta}$-field that exhibits relativity across users and contexts, and alignment through world synchronization. A distinctive property of RTG is *relativity*: different users or contexts align to different $\hat{\Delta}$ scales rather than collapsing into a single global notion of temporal distance. Figures 4 and Appendix C illustrate how users exhibit heterogeneous gap distributions, yet remain softly aligned through WS anchors. This contrasts with elapsed-time encodings, which impose the same scale across all users. By preserving user-dependent temporal coordinates, RTG enables structural alignment without erasing individual variability. These findings support treating temporality as an explicit structural inductive bias rather than leaving it implicit in hidden representations. Beyond these results, $\hat{\Delta}$ is not a lightweight

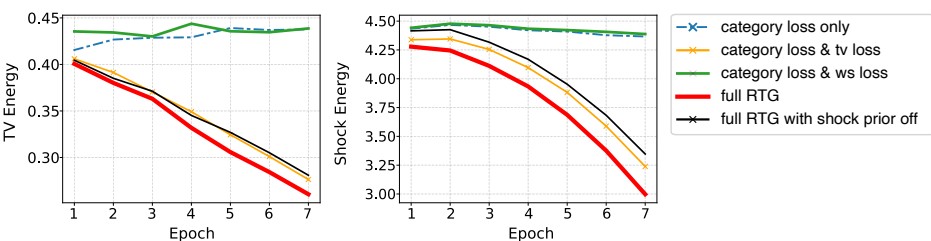

Figure 8: Field-level energy metrics across epochs (MIND). As expected, both energy decrease steadily when the TV prior is active, since the loss directly enforces local smoothness

Table 1: Multi-seed evaluation (seeds = 15, 42, 100). Results are reported as mean $\pm$ std. RTG outperforms sequential baselines, showing that the learned $\hat{\Delta}$-field carries predictive signal. Detailed ablation results are provided in the Appendix.

| Model | F1 | Precision | Recall |
|---|---|---|---|
| LSTM (sequence baseline) | $0.265 \pm 0.008$ | $0.186 \pm 0.025$ | $0.488 \pm 0.099$ |
| SASRec (Transformer baseline) | $0.312 \pm 0.007$ | $0.212 \pm 0.021$ | $0.626 \pm 0.140$ |
| MLP (feedforward baseline) | $0.344 \pm 0.006$ | $0.230 \pm 0.002$ | $0.681 \pm 0.028$ |
| RTG (this paper) | $0.390 \pm 0.020$ | $0.266 \pm 0.023$ | $0.741 \pm 0.038$ |

scalar like a timestamp; it is graph-valued, requires smoothing and anchoring, and depends on weak semantic priors for stability. Despite this complexity, it produces stable, user-specific distributions. Taken together, these properties suggest that $\hat{\Delta}$ can be interpreted as a semantic temporal coordinate rather than a direct proxy for $\Delta t$.

## 6.2 LIMITATIONS

Our comparisons are limited to classical sequential baselines (LSTM, SASRec), as the scope of this work is to test whether temporality can be modeled as structure. Incorporating stronger self-supervised and temporal GNN baselines is left for future work. In addition, the MIND dataset only provides row-level timestamps, preventing item-level supervision. We therefore restricted evaluation to field formation analysis. In practice this means RTG was tested under coarse and noisy supervision, where it still formed a stable $\hat{\Delta}$-field, suggesting that the inductive bias extends beyond precise elapsed-time conditioning.

## 7 CONCLUSION AND FUTURE WORK

We presented the Relational Time Graph (RTG), which encodes temporality as a learnable $\hat{\Delta}$-field on interaction graphs. Across commerce and news datasets, RTG forms a stable piecewise-smooth field that reflects relativity across users and alignment through world synchronization. These results demonstrate that temporality can be modeled as an explicit structural inductive bias, rather than left implicit in hidden representations.

Looking ahead, RTG opens several directions. Stronger baselines such as self-supervised sequential models and continuous-time GNNs will help clarify the comparative benefits of structural time. Datasets with item-level timestamps can enable finer supervision and sharper evaluation. More principled approaches to handling shocks may further broaden applicability. Beyond recommendation, we expect explicit temporal structure to be relevant for domains such as long-term memory modeling and concept drift detection, where temporal representation is central.

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

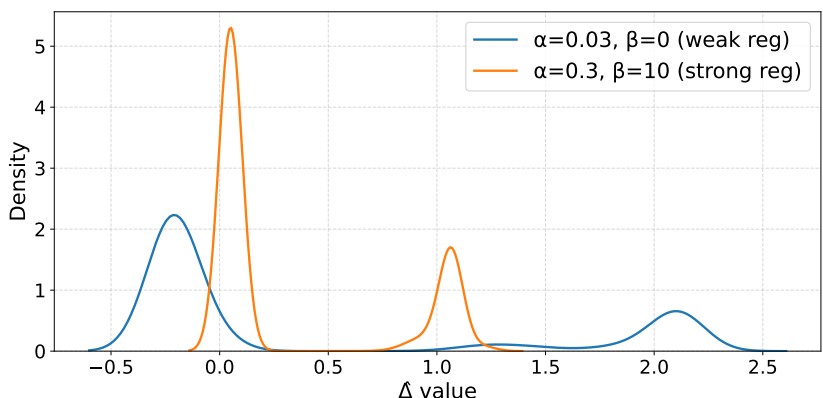

Figure 9: $\hat{\Delta}$ distributions under weak vs. strong regularization. Stronger $\alpha, \beta$ concentrate values more tightly, while weaker settings yield broader, less prominent modes.

Ashish Vaswani, Noam Shazeer, Niki Parmar, Jakob Uszkoreit, Llion Jones, Aidan N Gomez, Łukasz Kaiser, and Illia Polosukhin. Attention is all you need. *Advances in neural information processing systems*, 30, 2017.

Fangzhao Wu, Ying Qiao, Jiun-Hung Chen, Chuhan Wu, Tao Qi, Jianxun Lian, Danyang Liu, Xing Xie, Jianfeng Gao, Winnie Wu, et al. Mind: A large-scale dataset for news recommendation. In *Proceedings of the 58th annual meeting of the association for computational linguistics*, pp. 3597–3606, 2020.

Louis-Pascal Xhonneux, Meng Qu, and Jian Tang. Continuous graph neural networks. In *International conference on machine learning*, pp. 10432–10441. PMLR, 2020.

Jeffrey M Zacks, Nicole K Speer, Khena M Swallow, Todd S Braver, and Jeremy R Reynolds. Event perception: a mind-brain perspective. *Psychological bulletin*, 133(2):273, 2007.

# A  HYPERPARAMETER SENSITIVITY

## A.1  TV AND WS WEIGHT SWEEPS

### A.1.1  RETAIL ROCKET

We analyze the effect of TV and WS regularization weights ($\alpha$ and $\beta$ from Eq. (1)) on field formation. Figure 9 shows how the $\hat{\Delta}$ distribution tightens under stronger regularization, while Figures 10 quantify the effect on TV-energy.

Figure 10 (left) plots final TV-energy as $\alpha$ increases. When $\beta$ is near 0, TV plays a dominant role: larger $\alpha$ sharply reduces energy. When $\beta \geq 1$, the baseline energy is already low due to WS anchoring, so the marginal benefit of increasing $\alpha$ is smaller. This confirms that $\alpha$ is the direct knob for local smoothness, especially in the absence of world-synchronization.

Figure 11 reports predictive performance across a grid of $(\alpha, \beta)$ settings (seed = 42). Across the entire sweep, scores remain within $0.419$–$0.421$ except for a single dip at ($\alpha = 0.3, \beta = 3$) where $F1 = 0.407$. This plateau pattern shows that RTG is not sensitive to the weighting of TV and WS regularizers in terms of task performance. Hence $\alpha$ and $\beta$ should be interpreted not as hyperparameters for tuning accuracy, but as knobs for controlling field structure.

### A.1.2  MIND

We also examine the effect of $\alpha$ and $\beta$ on MIND. Figure 12 compares $\hat{\Delta}$ distributions under weak and strong regularization. Under weak regularization ($\alpha = 0.03, \beta = 0$), the distribution is broad

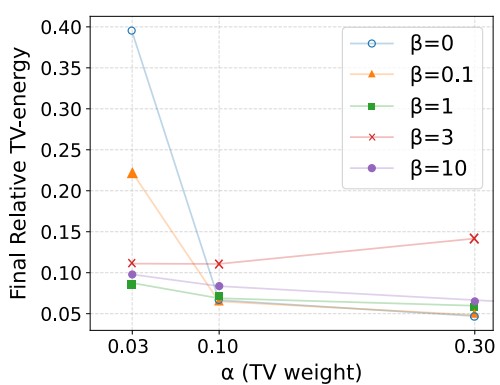

Figure 10: TV-energy vs. $\alpha$.

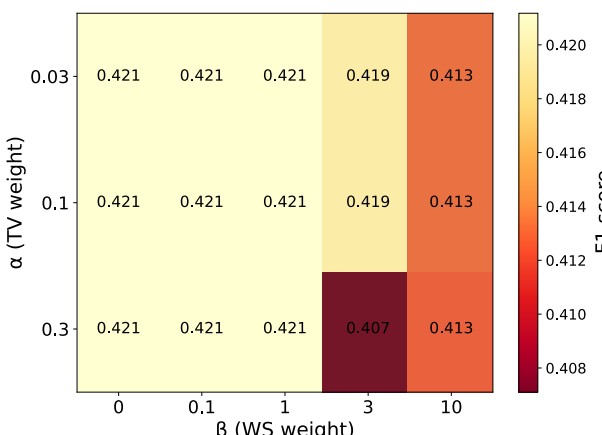

Figure 11: F1 score heatmap over $\alpha$ (TV weight) and $\beta$ (WS weight), seed $= 42$. Performance is flat across most of the grid (0.419–0.421), indicating robustness to hyperparameter choice.

with a dominant peak near $\hat{\Delta} \approx 4$ and smaller side bumps at lower values. With strong regularization ($\alpha = 0.3, \beta = 10$), the distribution concentrates around $\hat{\Delta} \approx 2$, reorganizing into a new single peak.

Figure 13 plots final TV-energy as $\alpha$ increases for different $\beta$ values. Across all settings, TV-energy decreases sharply with larger $\alpha$, confirming that $\alpha$ directly controls local smoothness. Unlike Retail Rocket, the role of $\beta$ is relatively minor: curves are nearly overlapping, showing that world-synchronization anchors provide only a small additional alignment signal on MIND. This is consistent with the weaker cross-user anchoring available in news sessions compared to e-commerce.

## B    EXTENDED ABLATION RESULTS (RETAIL ROCKET)

We report multi-seed performance (mean $\pm$ std over seeds {15, 42, 100}) for RTG variants on Retail Rocket. As anticipated from the main text, TV and WS act primarily as *training-time* priors: at evaluation, predictions depend on the learned $\hat{\Delta}$, so aggregate F1 stays flat across several ablations, while field-level metrics (distributions, TV-energy) diverge (see Fig. 3, Fig. 5 in the main text).

**Graph base only**   This setting shows that message passing alone provides improvements over classical sequence baselines (Table 1, main text), but its performance remains below RTG variants that incorporate a learned $\hat{\Delta}$ field.

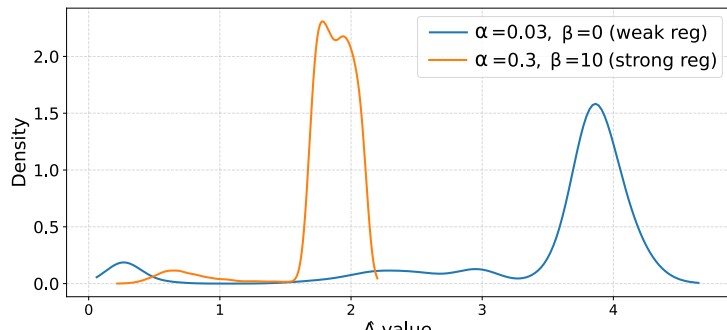

Figure 12: $\hat{\Delta}$ distributions on MIND under weak vs. strong regularization. Weak settings produce a broader, fragmented distribution, while stronger settings yield a narrower distribution with more prominent peak.

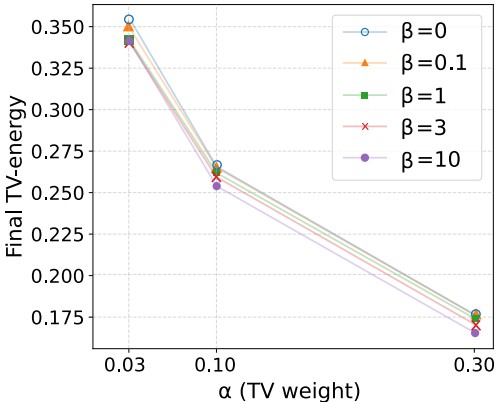

Figure 13: Final TV-energy on MIND as a function of $\alpha$ (TV weight) and $\beta$ (WS weight). Increasing $\alpha$ strongly reduces TV-energy across all $\beta$, while differences across $\beta$ remain small.

$\hat{\Delta}$ **only**  Achieving nearly the same F1 as the full model indicates that the $\hat{\Delta}$ predictor carries the primary predictive signal. The TV and WS components act mainly as training-time regularizers, shaping field structure rather than directly affecting accuracy.

**WS-shuffle**  Performance under WS-shuffle is comparable to the full model, which is consistent with the view that WS contributes mainly through item–bucket grouping. Shuffling user assignments preserves these anchor groups, resulting in similar aggregate accuracy.

**Full minus WS**  Removing WS maintains F1 at a similar level, but the learned field becomes less stable (higher TV-energy). This suggests that WS serves a structural role rather than a predictive one.

**Full minus TV**  Excluding TV leaves global performance largely unchanged, which is consistent with its local smoothing effect: differences appear in TV-energy rather than in aggregate F1. Note

that near-identical numbers across several rows reflect that these priors do not inject new predictive features; they shape the $\hat{\Delta}$-field (distributional shape, TV-energy, WS alignment) rather than headline accuracy.

Table 2: Retail Rocket ablations (mean $\pm$ std over seeds). Despite similar aggregate F1 across several settings, field structure differs markedly (see Fig. 3, Fig. 5 in the main text).

| Variant | F1 | Precision | Recall |
|---|---|---|---|
| Graph base only | $0.364 \pm 0.024$ | $0.236 \pm 0.023$ | $0.807 \pm 0.048$ |
| $\hat{\Delta}$ only | $0.392 \pm 0.025$ | $0.271 \pm 0.037$ | $0.734 \pm 0.080$ |
| RTG Full ($\hat{\Delta}$+Rank+TV+WS) | $0.390 \pm 0.020$ | $0.266 \pm 0.023$ | $0.741 \pm 0.038$ |
| RTG Full w/ WS-shuffle | $0.390 \pm 0.020$ | $0.266 \pm 0.023$ | $0.741 \pm 0.038$ |
| RTG Full minus TV | $0.390 \pm 0.020$ | $0.266 \pm 0.023$ | $0.741 \pm 0.038$ |
| RTG Full minus WS | $0.392 \pm 0.025$ | $0.272 \pm 0.036$ | $0.719 \pm 0.070$ |

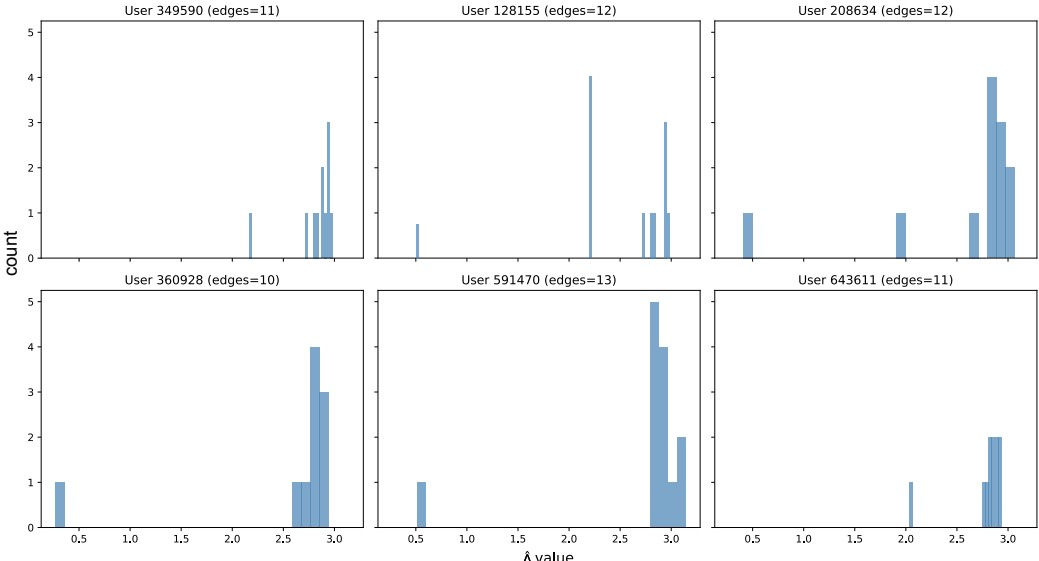

Figure 14: User-specific $\hat{\Delta}$ distributions on MIND. Different users align to different gap scales, highlighting relativity: temporal distances are learned in a user-dependent manner rather than by a single global scale.

## C   USER-SPECIFIC $\hat{\Delta}$ DISTRIBUTIONS (MIND)

Figure 14 shows histograms of $\hat{\Delta}$ for six randomly selected users (each with $\geq 10$ edges). The distributions differ markedly across users: some concentrate near small gaps, others peak at larger values, and the spread also varies. This illustrates the *relativity* property of RTG: the learned $\hat{\Delta}$-field does not enforce a single global notion of temporal distance, but adapts scales to each user's interaction pattern. In other words, event gaps are represented in a user-dependent coordinate system, while remaining structurally aligned through WS anchors.

## D   MEAN $\hat{\Delta}$ BY TRANSITION TYPE (RETAIL ROCKET)

To better interpret the bimodal distribution in Figure 3, we analyze average $\hat{\Delta}$ values grouped by transition type on Retail Rocket (1=view, 2=add-to-cart, 3=transaction). As shown in Figure 15, transitions that end in *view* actions (e.g., $1\to1$, $0\to1$, $2\to1$, $3\to1$) cluster near zero, while transitions that end in *add-to-cart* or *purchase* (e.g., $1\to2$, $1\to3$, $2\to2$, $2\to3$) take substantially larger values, close to 1.0 or higher. This confirms that the lower peak near 0.0 in Figure 3 primarily captures transitions to *view*, while the higher-density region near 1.0 corresponds to transitions leading to purchase-oriented actions. Together, these results demonstrate that the learned $\hat{\Delta}$ field meaningfully separates exploratory versus goal-directed user behaviors.

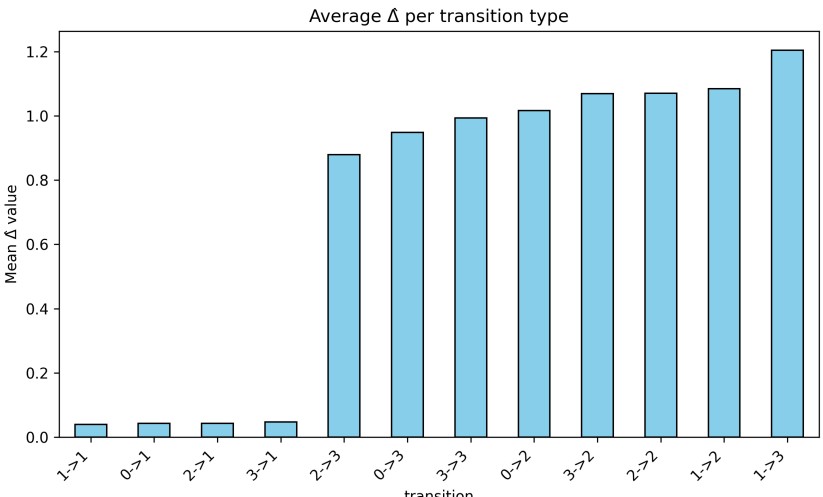

Figure 15: Mean $\hat{\Delta}$ per transition type on Retail Rocket. Transitions ending in *view* cluster near 0, while transitions ending in *add-to-cart* or *purchase* show substantially larger values.

# E   IMPLEMENTATION DETAILS

**Runtime.**   Retail Rocket experiments were run on a MacBook Air CPU and completed within 20 minutes including preprocessing. MIND-Large experiments were run on a Lambda Labs instance with NVIDIA A100 GPUs and completed within about 1 hour including preprocessing.

## REPRODUCIBILITY STATEMENT

We provide dataset descriptions, preprocessing steps, model components, and hyperparameters in Section 4 and Appendix D. Unless otherwise stated, experiments were run with random seed $42$. For Retail Rocket, Table 1 and Table 2 report averages over seeds $\{15, 42, 100\}$. Default regularizer weights were $(\alpha, \beta) = (0.1, 10.0)$ for Retail Rocket and $(\alpha, \beta) = (0.1, 0.1)$ for MIND-Large. Runtime environments (CPU and GPU) and training times are reported in Appendix D. Code and preprocessing scripts will be released upon publication.

## USE OF LLMS

Large language models were used in the preparation of this paper. Specifically: (1) to aid grammar checking and tone polishing during writing, and (2) to help discover and verify recent related work.

