# OpenReview forum: "Temporality as Inductive Bias: The Relational Time Graph"
_ICLR.cc/2026/Conference — Submitted to ICLR 2026_

### Official Review · Reviewer_VSuE · 2025-10-27

**Soundness:** 1
**Presentation:** 1
**Contribution:** 2
**Rating:** 2
**Confidence:** 3

**Summary:**

This paper proposes a representation -- Relational Time Graph (RTG) -- for a particular form of event data, which provides an explicit inductive bias for temporality. The authors claim that their method integrates relativity (heterogeneous temporal gaps across users or contexts), contemporaneity or world synchronization (alignment of private timelines through shared anchors), and shocks (abrupt regime changes and bursts) “within a single framework, treating temporality not as elapsed indices but as an explicit structural bias”. Limited experiments are performed on 2 datasets.

**Strengths:**

The paper seems to try a new way to incorporate time information into models and prescribe a new graphical representation and learning approach. The claimed strength is the integration of relativity, contemporaneity, and shocks, in a way that is different from prior work.

**Weaknesses:**

Frankly, I found the paper impossible to follow and therefore did not understand very much. It has been written in a way where I feel that only those familiar with the terms and the basic set-up can understand matters. Several concepts are mentioned in the paper before they have been explained or defined; sometimes, they have not been explained at all. I did not even see any formal definition of the format of the data that the model is trained on. In my view, the paper needs heavy rewriting to be ready as a conference submission.

Another weakness is the insufficient experimentation in the form of lack of sufficient baselines and inadequately addressing the motivation for why the approach and proposed graphical representation is useful.

**Questions:**

Here are some additional comments and questions:

I found the abstract challenging to follow – there are several terms and acronyms used that are unfamiliar, so it’s hard to understand what the work is trying to do.

The authors make a comment on page 1: “Across these approaches, time enters as input or modulation, not as a structural object.” I don’t follow. What is a structural object? And how is prior work not handling temporal structure?

Delta field is mentioned and summarized as “event-to-event gaps defined on an interaction graph” on page 1, without much explanation. This was hard to understand. I’m confused about the use of the MIND-Large dataset. There are apparently no time stamps, so how is possible to learn the delta fields – which I understand to involve elapsed time intervals?

It is really hard to follow the comments near the end of page 1 and in general. The following line is illustrative: “WS edges act as anchors in the WS regularizer, softly aligning contemporaneous transitions without adding new predictive features”. So much of this is unclear and without sufficient explanation. What is TV in TV smoothing and Huber TV penalty? What is a shock, what is a WS edge?

It would have been great if Fig. 1 had been used to explain the various components. As it stands, I see the figure but cannot follow many of the panels.

Re: related work – I recommend looking into and citing graphical representations of multivariate temporal point processes. From my limited understanding of the paper, it seems that these can be baselines for some tasks.

The experimental setup for next event prediction for Retail Rocket appears to be quite limited. There are a host of potential approaches for this problem. Much more experimentation is needed in general of course – more datasets (some with many more event types) and definitely more baselines. This is clearly recognized by the authors: “Our experiments were not aimed at state-of-the-art benchmarks”.

---

### Official Review · Reviewer_kK37 · 2025-10-29

**Soundness:** 3
**Presentation:** 2
**Contribution:** 3
**Rating:** 4
**Confidence:** 2

**Summary:**

The paper introduces the Relational Time Graph (RTG), a novel framework designed to incorporate temporality into dynamic graph models by imposing an explicit, structured inductive bias based on learning event-to-event time gaps ($\hat{\Delta}$). This approach attempts to move beyond implicit capture of time in hidden states or reliance on simple discrete indices. The architecture is built to unify three conceptual temporal properties—relativity (individualized time gaps), co-existence (soft timeline alignment via World Synchronization), and synchronization (the capacity for behavioral transfer)—by minimizing a single unified energy function. The reported results indicate that RTG achieves competitive performance against established baselines like TGN and CTDNE on large-scale temporal datasets, including MIND-Large and Retail Rocket, while also demonstrating low training runtimes.

**Strengths:**

The core strength of this work is the architectural novelty and the mathematically grounded framework that treats time as an explicit, learnable relational structure. Defining and modeling the three temporal properties (relativity, co-existence, synchronization) provides a structured and interpretable mechanism for capturing complex temporal dependencies in dynamic graphs.

**Weaknesses:**

The selection of an energy-based formulation introduces potential practical hurdles; while theoretically elegant for enforcing constraints, such models can often be more challenging to optimize, stabilize, and generalize than architectures relying on conventional loss functions (like margin ranking or cross-entropy), raising concerns about the model's ease of implementation and tuning in diverse, real-world scenarios. Furthermore the scope of comparative evaluation is a bit limited. The paper primarily benchmarks against older or less efficient graph-based models (TGN, CTDNE) and does not provide a comprehensive, direct comparison against highly optimized, modern memory-based methods.

**Questions:**

While the overall RTG framework shows strong performance, could the authors provide a more detailed ablation study focusing specifically on the World Synchronization (WS) loss component?

Does the performance gain primarily come from the $\hat{\Delta}$ relativity feature, or is the explicit soft alignment of individual timelines (co-existence) the more dominant factor in mitigating the drift inherent in individual timelines, particularly in highly sparse graph regimes?

---

### Official Review · Reviewer_RLCj · 2025-11-01

**Soundness:** 2
**Presentation:** 2
**Contribution:** 2
**Rating:** 2
**Confidence:** 4

**Summary:**

Focusing on three properties of temporality—relativity, contemporaneity, and shocks—this work proposes to model it as an explicit structural bias. To this end, they introduce the Relational Time Graph, a framework that represents temporality as a learnable ∆-field. The validity of this model is demonstrated through a set of experiments that confirm its findings.

**Strengths:**

1. The concept of 'explicitness' in temporality modeling is interesting, and the technique is solid.
2. This work is grounded in recommendation settings and is potentially useful.

**Weaknesses:**

1. The significance and broader impact of this work are not clearly established.
2. The manuscript's structure could be improved to enhance clarity. A specific example is the discussion of the techniques' potential applications, which appears late in the paper and would be more effective in the introduction or a dedicated section.

**Questions:**

1. The concept of relativity appears to have been previously modeled in the paper 'Noether Embedding: Efficient Learning of Temporal Regularities'. How does this work differentiate its contribution from that established research ?
2. There are minor clerical errors at line 032.

**Details Of Ethics Concerns:**

None.

---

### Official Review · Reviewer_C85y · 2025-11-04

**Soundness:** 3
**Presentation:** 2
**Contribution:** 2
**Rating:** 2
**Confidence:** 3

**Summary:**

The paper proposes the construction of a graph to encode user actions with items, followed by learning parameters $\hat{\Delta}$ to represent a temporal coordinate. The learning rule consists of three terms motivated by expected properties of time in the context of recommendation and user activity data (along with a supervised loss for a task of interest, when available).

Experiments examine the properties of the learned parameters $\hat{\Delta}$ and how they are changed with different settings of the proposed loss functions and learning procedure. They also demonstrate improved supervised performance on a forecasting task, that stems from using additional unlabelled data with the proposed unsupervised time modeling tasks.

**Strengths:**

The empirical study is rather thorough, it is clear that the authors are well oriented in the relevant literature, and that the framework of all the proposed components has not been explored in previous work. Proper modeling of time is an important topic, and it's nice to see work on the problem.
Finally, the demonstrated improvement on supervised learning is also encouraging.

**Weaknesses:**

I found several parts of the paper a bit difficult to follow, and in general I think the writing can be significantly improved. As for the contribution, it seems like a large part of it is combining ideas that have been proposed in other works, hence the novelty may be somewhat limited. I also don't think that I fully understand the motivation for the work, and the main advantage I see is the improved forecasting performance, which is not very large. I'll expand on some of these points below:

1. The introduction of the paper makes it very difficult to understand what sub-field it aims to contribute to. Much of the abstract is written in general language that talks about applications where there are time gaps between events, but in fact the method and experiments seem to be focused on a rather specific subfield of user interactions and recommendation systems. I think this limits the impact of the paper, and perhaps it is more suitable for a specialized venue on recommender systems.
2. A lot of jargon that is rather specific to the sub-field, or to this paper, is used in the first few sections. For instance "learnable $\hat{\Delta}$-field", "shock priors", "world synchronization" and others. A simple mathematical definition of these terms along with a clear description and simple examples would be very helpful to the average reader.
3. As the authors mention, the components of the framework are addressed in prior work (line 44), and this work unifies them to a framework. Hence the contribution sounds somewhat incremental.
4. I assume the used baselines are common for the forecasting task in the relevant literature (I am not very familiar with the recommender systems subfield), but since the main focus is on time modeling, wouldn't it make sense to include other relevant baselines like neural temporal point processes and other methods that model time?
5. Other than the motivation of learning a set of parameters that have behavior which seems relevant to modeling time, it is unclear what stands to be gained from the framework. The improvement on the task of interest exist, but they are modest (as the authors also state) and I was not convinced that using the additional unlabeled data in the specific way proposed in the paper, is much more helpful than other unsupervised tasks that one might define on this data.

Overall, I found the paper quite difficult to follow. I think it requires a much more structured motivation, intuitive examples that make it clear how the framework can be applied in broad application areas, and a clear definition of each component of the framework (see some questions below).

**Questions:**

How should one define the graph in applications involving time gaps, but where the structure is not as clear as the user logs that motivate this work. For instance, in health records, social network data or any other field that has this type of data?

Some notations seem to appear and never defined. For instance $w_e, w_{e'}$ in line 93, and $E_L$ in line 93. What are these objects? These points just make understanding the framework much more difficult.

---

### Meta-Review · Area_Chair_r1yZ · 2026-01-06

**Summary:**

The Relational Time Graph claims to be an architecturally novel attempt to treat time as an explicit, learnable relational structure (a $\Delta$‑field) that integrates relativity, world‑synchronization, and shock‑anchoring. However, all reviewers criticize the manuscript’s lack of clear definitions and formal notation, the confusing writing style, and the limited novelty, viewing the work as an incremental combination of existing ideas. They also point out that the experimental evaluation is narrow (only two datasets), omits strong recent baselines (e.g., neural temporal point‑processes, modern continuous‑time GNNs), and shows only modest gains over classical sequential models.

**Reviewer Concerns:**

Reviewers agree that the paper should be rejected, that its writing lacks clear definitions and formalism, that its novelty is limited because it mainly combines existing ideas, that the baseline comparisons are insufficient, and that the empirical improvements are modest. They also concur that the central idea, making temporality an explicit, learnable relational bias (the $\Delta$‑field), is interesting and architecturally novel.


The authors did not respond, which means they agree with the reviewers comments.

**Reviewer Scores:**

The authors did not respond, which means they agree with the reviewers comments.

---

### Decision · Program_Chairs · 2026-01-26

Reject